# Characterization of a wind turbine wake evolving over an intertidal zone performed with dual-lidar observations

Changzhong Feng<sup>1</sup>, Bingyi Liu<sup>1</sup>, Songhua Wu<sup>1,2</sup>, Jintao Liu<sup>1</sup>, Rongzhong Li<sup>3</sup>, Xitao Wang<sup>3</sup>

<sup>1</sup>Ocean Remote Sensing Institute, Ocean University of China, Qingdao 266100, China

<sup>5</sup> <sup>2</sup>Laboratory for Regional Oceanography and Numerical Modeling, Qingdao National Laboratory for Marine Science and Technology

<sup>3</sup>Seaglet Environmental Technology, Qingdao, China

Correspondence to: (liubingyi@ouc.edu.cn)

Abstract. As modern wind power industry quickly develops, it is of high priority to optimize layouts and operations of wind turbines to reduce the influences of wakes induced by upstream wind turbines. The wake behaves complicatedly with landocean-atmosphere interactions. This complex wake could be observed by two or more synchronously operated Doppler lidars. Accordingly, we characterized a wind turbine wake evolving over an intertidal zone performed with dual-lidar observations. Dynamic process of wakes merging that occurred from approximately 1 D (rotor diameter) downstream was captured and analysed. The phenomenon that wake length increased with rising tide was analysed in details. It suggested that

15 the increase of wake length varied with underlying surface roughness transition from mud to sea water as well as the rising sea level. Finally, wake meandering cases were analyzed in detail. Our research shows that the dual-lidar observation technology is a promising remote sensing tool for characterization of complicated wind turbine wakes.

#### **1** Introduction

30

Wind energy is no doubt one of the most promising alternative energies. As wind power industry increases rapidly during the

- 20 past decades, optimizations of wind turbines layouts and operations have presented a significant challenge to minimize the cost of energy. Specifically, considerably decreased output power and enhanced fatigue are induced by upstream turbine wakes (Baker and Walker, 1984; Barthelmie et al., 2003; Chowdhury et al., 2012). Therefore, various experiments and simulations have been performed previously in an attempt to investigate the dependence of the wake behaviours (velocity deficit, wake length, wake boundary and wake centreline) on the atmospheric conditions (wind speed, turbulence, surface
- 25 roughness and atmospheric stability). Full-scale experiment could convincingly help the validation of large-eddy simulation (LES) technique combined with turbine model (Abkar and Port éAgel, 2015; Iungo et al., 2013; Port éAgel et al., 2011; Wu and Port éAgel, 2012), and ultimately improve the prediction of wind power harvesting (Aitken and Lundquist, 2014; Fuertes et al., 2014; Hirth et al., 2012).

The conventional approach to obtain wind data at a fixed point is to utilize cup anemometers mounted on meteorological towers (Baker and Walker, 1984; Elliott and Barnard, 1990). However, the investigation of complex wake structure needs

more data points. For instance, Elliott et al. analysed wake characteristics from wind data of nine meteorological towers (Elliott and Barnard, 1990). Nevertheless, the impact of towers with large cross-sections on the measurements at hub height should be taken into consideration. Therefore, the remote sensing technique such as lidar, radar, and sodar has been increasingly adopted in experimental turbine wake research. Högström et al. utilized four different equipments, including a

- kite anemometer, a tower-mounted instrumentation, a tethered balloon sounding and a Sonic Detection and Ranging (SODAR), to obtain the vertical profile of turbulence intensity (Högström et al., 1988). Kambezidis et al. employed three SODARs placed at equal distances downstream of a turbine to investigate velocity deficit, turbulence intensity and temperature structure within the wake (Kambezidis et al., 1990). In situ measurement combine with acoustic sounders was used to identify rotational motion inside the wake by Helmis et al. (Helmis et al., 1995). Barthelmie et al. elaborately
- evaluated the operation of the SODAR mounted on a ship for measuring offshore wind turbine wake, and quantified the relationship between wind farm efficiency and various atmospheric conditions (Barthelmie et al., 2003; Barthelmie and Jensen, 2010).

For a pulsed coherent Doppler lidar (PCDL), access to the full-scale wind turbine wake could be provided by a variety of geometrical scanning modes (Bing öl et al., 2010; K ösler et al., 2010; Trujillo et al., 2011; Wu et al., 2016; Wu et al., 2014).

- The lidar deployment and wind direction should be taken into account because lidar directly obtains Light of Sight (LOS) velocity. For instance, the lidar-turbine line should be strictly aligned with the turbine wake in the RHI (Range Height Indicator) mode (Kopp et al., 2004; Smalikho et al., 2005), which sweeps elevation angle with a fixed azimuth angle (K äsler et al., 2010). However, this scanning mode could not always intersect the wake particularly at far wake region due to wake meandering and the variation of wind direction and turbine yaw. Similarly, when lidar operates by sweeping azimuth angle
- with the constant elevation angle, called Plane Position Indicator (PPI) mode, the distance and orientation from lidar to turbine should be sufficiently far and roughly same with wind direction (Smalikho et al., 2013), respectively. In general, detected wake length is limited by the increasing altitude with range and wake orientation for both PPI and RHI mode of PCDL. Furthermore, analysis of these measurements is based on LOS velocity or simple projection along wake orientation. Vector wind field could be retrieved based on two or more independent measurements (Armijo, 1969; Ray et al., 1978;
- Rothermel et al., 1985). Three pulsed coherent lidars of Leosphere were employed to deduce the measurements at a fixed point, which was compared with the data collocated from sonic anemometer by Mann et al. (Mann et al., 2009). To investigate axial and vertical velocity components, simultaneous measurements utilizing two lidars with RHI mode were performed behind a wind turbine by Iungo et al. (Iungo et al., 2013). For this case, one lidar was placed at the turbine location and the other one was located downstream. Hirth et al. employed two radars to reconstract wake structure, and then
- discussed the variability of a single turbine wake and the complex flow features included in a wind farm (Hirth and Schroeder, 2013; Hirth et al., 2015).

#### 1.1 Velocity deficit and wake length

Definition of velocity deficit at hub height in previous studies can be written as:

 $\delta(x) = \frac{U_{ref}(x) - U_{wake}(x)}{U_{ref}(x)} \times 100\% ,$ 

where,  $U_{ref}(x)$  is the ambient or reference velocity as a function of longitudinal distance x downstream from the wind turbine,  $U_{wake}(x)$  is the wake velocity. The initial value depends on the amount of momentum loss induced by wind turbine (Wu et al., 2016), with typical value of approximately 50%-60% (Aitken et al., 2014), 66% in (Kambezidis et al., 1990) and

74% in (Smalikho et al., 2013). Wake length L is the distance, at which the velocity deficit drops down to 10% (Smalikho et 5 al., 2013), with typical value of 7–10 rotor diameters (D) and extreme value of exceeding 20 D (Hirth and Schroeder, 2013) and 30 D (Hirth et al., 2012).

The environmental factors that affect velocity and wake length include wind speed, turbulence, atmospheric stability and surface roughness. Researches indicate that velocity deficit is notably higher at lower ambient velocity owing to higher thrust

- 10 coefficients (Elliott and Barnard, 1990). It shows that the lowest turbulence intensity occurs at 8–12ms<sup>-1</sup>, in which case the velocity deficit is high due to high turbine thrust coefficient (Elliott and Barnard, 1990; Hansen et al., 2012). Besides, turbulence plays an important role in mixing effect between wake and surrounding air (Aitken et al., 2014), and strongly affects velocity deficit in the far wake (Rhodes and Lundquist, 2013). For instance, Smalikho et al. proposed that wake length would be half when the turbulent energy dissipation rate doubles at high wind speed (Smalikho et al., 2013). In
- addition, unstable atmospheric conditions would result in the enhancement of turbulence intensity levels. Such as, wind 15 turbine wake recovers faster in convective condition than in neutral condition (Iungo, 2016). However, the factor of surface roughness on wind farm needs more elaborate investigations.

## 1.2 Wake centreline and width

As pointed out by Vermeer et al., downstream wake could be separated into two parts: near and far wake regions, with the dividing line a few D downwind of the corresponding wind turbine (Vermeer et al., 2003). In the near wake region, the 20 velocity behind the turbine rotor decreases owing to the lift generated along the blade, with a maximum lift value near 75% blade span. Consequently, wind profile of horizontal cross section in the near wake region has two peaks (Smedman, 1998), or double-bell shape (Aitken et al., 2014; Larsen et al., 2007),

$$U(y,r) = U_{ref} - a_1 \times \left\{ \exp\left[ -\left(\frac{y - y_1}{c_1}\right)^2 \right] + \exp\left[ -\left(\frac{y - y_2}{c_1}\right)^2 \right] \right\},$$
(2)

As distance increases downwind, tip vortexes with spirals pattern gradually entwine into one with cross section wind profile 25 of single centreline peak or Gaussian-like shape (Aitken et al., 2014; Larsen et al., 2007),

$$U(y,r) = U_{ref} - a_0 \times \exp\left[-\left(\frac{y-y_0}{c_0}\right)^2\right].$$
(3)

In Eq.(2) and Eq.(3)  $U_{ref}$  is the ambient wind speed,  $a_1$  and  $a_0$  are the amplitudes of the Gaussian curve,  $C_1$  and  $C_0$  are the parameters to describe the wake width,  $y_1$ ,  $y_2$  and  $y_0$  are the locations of the left, right and center minima in the double-

(1)

Gaussian and single-Gaussian curves, respectively. As defined in (Aitken et al., 2014; Hansen et al., 2012), wake width of single-Gaussian shape, with definition of 95% confidence interval of velocity profile, can be written as:

$$w_1 = 2\sqrt{2}C_0$$

while, width of wake with double-Gaussian distribution can be expressed as:

$$w_2 = 2\sqrt{2}C_1 + |y_1 - y_2|$$
, (5)

As wake propagates downstream, wake width and centreline behave as in what follows. Due to the mixing effect by smallscale atmospheric eddies (Bingöl et al., 2010), wake expands larger with incremental distance downwind in the horizontal direction than in vertical direction owing to ground effect (Aitken et al., 2014). In addition to wake width, wake centreline shifts upward in the vertical direction due to the larger momentum at the lower part of the wake than in the higher part of the

10 wake (Helmis et al., 1995) and the tilt of the rotor (Aitken et al., 2014). Besides, horizontal displacements of wake centreline are larger than vertical displacements with the ratio of approximately 1.5 (España et al., 2011).

This paper presents characterization of intertidal turbine wakes based on the two pulsed coherent Doppler lidars. In what follows, section 2 subsequently describes the dual-Doppler experimental setup. In section 3, observation results, including wakes merging, wake evolution with rising tide and wake meandering, are presented and analysed. Finally, summary and discussion are offered in section 4.

#### 2 Description of the experiment

A dual-lidar experiment for measuring wind turbine wakes was performed on December 15, 2014, with cloudy weather condition and westerly to northwesterly wind at the Jiangsu Rudong intertidal wind farm ( 32 31'17" N, 121 °10'32" E). Two lidars were deployed on the embankment with the intertidal zone at the top right of the lidar-lidar line and the coast at the

- bottom of Fig. 1. The tilted plane scanning mode was adopted for turbine wakes observation with elevation denoted by the gray contour lines in Fig. 1. The 2-MW wind turbine T1 had a 100-m rotor diamter and a hub height of 80m. The hub height and rotor diamer of the 1.5-MW wind turbine T5, T6 and T7 were 80 m and 93 m, respectively. Two gasoline engine generators were deployed for power supply. Sea Surface Temperature (SST) from European Centre For Medium Range Weather Forecasts (ECMWF) showed that the sea surface temperature was appximatly 10 °which was at least 4 °higher than
- the air temperature (-1 °-6 ° at local weather station http://lishi.tianqi.com/rudong/201412.html). To obtain the rough time when tide began to rise and reached highest point, the tidal level data was achieved from the nearest Yangkou Harbour (<u>http://www.chinaports.com/chaoxi</u>), located southeast with a distance of approximately 18 km. Data from the harbour showed that tide began to rise after 1204 local time (LT) with tide height of 222 cm and reached the maximum value of 398 cm at 1811 LT. Accordingly, the tide at the experiment was supposed to be rising from 1300LT to 1700LT.