# Peer review of "Characterization of a wind turbine wake evolving over an intertidal zone performed with dual-lidar observations"

_Atmospheric Measurement Techniques, 2017_

## Referee Comment (RC1) · Anonymous Referee #2 · 27 Apr 2017

My final comment to the editor is that the 1-day data period seems to me too short for drawing firm conclusions. Furthermore, in my opinion no clear scientific questions are addressed with this paper.

---

## Referee Comment (RC2) · Anonymous Referee #1 · 29 Apr 2017

Overall Comment:

The manuscript describes the investigation of wake characteristics during a time period of rising tide. The goal of the paper is to determine if the rising tide contributes to changes in wake length. The topic of research is quite interesting and is worthy of study. However, there are several flaws in this manuscript. The current experiment design although unique, seems quite ill suited for the purpose of this study. In addition, the reviewer finds cases of improper logic and justification. With this in mind, the reviewer recommends a rejection of this paper. Hopefully the comments below will help the authors to refine their future plans for this study. Detailed comments and suggestions to the authors are given below.

[Figure]

Specific comments:

1) Why was the angle of the tilted plane chosen to be 4o? This angle seems quite steep to account for lifting of the wake center-line. In addition, the choice of only 1 scanning plane is poor experiment design as there is no guarantee that the wake will be aligned with this tilted plane.

2) The authors use a unique scanning geometry to perform the measurements. However, it is not clear, if this actually helps in the present case. One thing is clear, more measurement levels are required.

3) In terms of performing dual-Doppler, the accuracy of the retrieval is a function of the $\triangle AZ$ within each measurement volume. That is, the retrieval is more accurate as $\triangle AZ$ approaches 90 deg and less accurate as $\triangle AZ$ approaches 0 deg (or 180 deg). From the experiment design, it seems like the $\triangle AZ$ will be quite low towards the left and the right edges as well as close to the lidar locations. Therefore, to put the quality of the measurements in context, please include a figure showing the $\triangle AZ$ as well as follow uncertainty quantification as described in Simley et al. (2016).

4) There is no description of the data quality control. This should be properly defined.

5) Wake merging: From figures 2 and 3, panels (b), (c) and (d): It seems like the magnitude of the wake deficit is on the same order of the spatial variability within the individual transects. This is quite interesting and needs investigation. However, it is not the best example to study wake merging as several other background effects dominate.

6) As the authors point out themselves on page 8, line 20, the reduction in deficit is most probably due to the measurement plane leaving the wake region. Therefore, the "measured" wake length is not an accurate estimation of the actual wake length (the sudden drop in deficit should point to this). This (again!) points to the requirement of having several levels of measurements in order to accurately estimate the actual wake length.

7) The authors report that as the tide rises, so does the wake length. However, they fail to note the importance of wind direction. During the time period of the tide rise, the wind direction shifts from south-westerly to westerly. As the wind direction shifts to westerly, now, the wake from turbine T1 is measured. In this case, the measurement plane is almost parallel to the turbine hub axis along the wake direction. Hence, the wake region remains in the measurement plane for much longer, resulting in reporting of longer wake lengths. Therefore, the increase in wake length is NOT due to tide levels, but rather the angle of the measurement plane relative to the turbine hub axis! It just turns out that the wind direction shift is correlated with the rising tide and the authors mistake this correlation for causation.

8) Apart from the above point, any conclusions about the impact of change in surface roughness characteristics on wake length need to be back by reproducible results spanning several time periods. One case study is not enough as presented here. It is suggested to have several periods of wake measurements for each set of turbines with similar characteristics (hub-height, rotor diameter, wake fetch etc).

References:

Simley, E., Angelou, N., Mikkelsen, T., Sjöholm, M., Mann, J., and Pao, L. Y.: Characterization of wind velocities in the upstream induction zone of a wind turbine using scanning continuous-wave lidars, J. of Renewable and Sustainable Energy, 8, 013 301, 2016.
* * *

---

## Author Comment (AC1) · 8 May 2017

We have modified the conclusions and added the methodology and the description of the scanning in a new section. We thanks for your constructive advice. In your opinion, perhaps our paper is not so good. But we have been trying to make it better and we are pleasured to share our ideas with others.

Please also note the supplement to this comment:
http://www.atmos-meas-tech-discuss.net/amt-2017-23/amt-2017-23-AC1-supplement.pdf

[Figure]

[Figure]

**Supplement:**

**Characterization of a wind turbine wake evolving over an intertidal zone performed with dual-lidar observations**

Changzhong Feng[1], Bingyi Liu[1], Songhua Wu[1,2], Jintao Liu[1], Rongzhong Li[3], Xitao Wang[3]

[revised manuscript text omitted]

15 coplanar scanning mode, which can be used for ground-based measurements in condition of various wind directions. In what follows, section 2 describes the methodology of dual-Doppler method and coplanar scanning mode. A coplanar scanning mode to exactly match probing volume is presented in section 3 and applied in field experiments in December, 2014 in section 4. In section 5, observation results, including wakes merging, wake evolution with rising tide and wake meandering, are presented and analysed. Finally, summary and discussion are offered in section 5.

20 **2 Methodology**

**2.1 Principle of dual-Doppler method**

If two or more LOS velocities of a certain probing volume are synchronously detected by independent instruments from different directions, the vector wind at the certain probing volume is possibly derived without assumptions of homogeneous wind field. The principle of dual-Doppler method is described in this section. The geometrical relationship of measuring

25 vector wind (*V*) at a certain probing volume (denoted by point *C*) with two instruments (denoted by points *A* and *B*) is shown in Fig. 1. The detected vertical wind component is neglectable compared with horizontal wind because the detected LOS projection of vertical wind is very small when the transmitting beam emits nearly horizontally. Therefore, for simplification only horizontal wind is considered.

[Figure]

**Figure. 1. Geometrical relationship of measuring vector wind (denoted by $V$) at a certain probing volume (denoted by point $C$) with two instruments (denoted by points $A$ and $B$) using dual-Doppler method. $V_A$ and $V_B$ are LOS components of vector wind $V$ measured by instruments $A$ and $B$, respectively. $\alpha$ and $\beta$ are angles between north and radial direction of instruments $A$ and $B$, respectively.**

As shown in Fig. 1, the position of instrument A (point $A$) is defined as the origin of geographic coordinate system. X-axis pointing in east indicates zonal direction, while Y-axis pointing in north indicates meridional direction. $\alpha$ and $\beta$ are angles between north and radial direction of instruments A and B, respectively. $V_A$ ($u_A$, $v_A$) and $V_B$ ($u_B$, $v_B$) are LOS components of vector wind $V$ ($u$, $v$) measured by instruments A and B, respectively, which can be expressed as

$$
\begin{cases}
u_A = V_A \sin \alpha \\
v_A = V_A \cos \alpha \\
u_B = V_B \sin \beta \\
v_B = V_B \cos \beta
\end{cases}
\tag{6}
$$

$V'_A$ ($V'_B$) is the difference between $V$ and $V_A$ ($V_B$). Since $V'_A$ and $V'_B$ are perpendicular to $V_A$ and $V_B$, respectively, Eq. (7) can be obtained as

$$
\begin{cases}
u_A \cdot (u - u_A) + v_A \cdot (v - v_A) = 0 \\
u_B \cdot (u - u_B) + v_B \cdot (v - v_B) = 0
\end{cases}
\tag{7}
$$

The vector wind $V$ ($u$, $v$) at point $C$ can be derived from Eq. (6) and Eq. (7) as

$$
\begin{cases}
u = \dfrac{V_A \cos \beta - V_B \cos \alpha}{\sin(\alpha - \beta)} \\
v = \dfrac{V_B \sin \alpha - V_A \sin \beta}{\sin(\alpha - \beta)}
\end{cases}
\tag{8}
$$

Subsequently, wind speed $V$ and wind direction $\theta$ can be obtained as:

$$
V = \sqrt{u^2 + v^2} ,
\tag{9}
$$

$$\theta = \begin{cases} \dfrac{\pi}{2} - \arctan\left(\dfrac{v}{u}\right), & u > 0 \\[3mm] \dfrac{3\pi}{2} - \arctan\left(\dfrac{v}{u}\right), & u < 0 \end{cases}$$

(10)

After scanning measurements are synchronously performed by two instruments, vector wind filed in overlapped area could be derived without assumption.

**2.2 Estimation of wind uncertainty**

5    In order to evaluate the performance of the measurements based on dual-Doppler method, an estimation of wind uncertainty is presented. According to Eq. (3), the $u$ and $v$ components of derived vector wind $V$ are functions of four variables, $\alpha$, $\beta$, $V_A$ and $V_B$, as a result the wind uncertainty is contributed from the uncertainty of these four variables. Assuming the pointing error of both instruments is very small during the scanning measurements, the uncertainty of $\alpha$ and $\beta$ can be neglected. Considering $V_A$ and $V_B$ are measured by different instruments, the uncertainty of $V_A$ and $V_B$ should be independent. Therefore,

10    the uncertainty of $u$ and $v$ components, denoted as $\delta u$ and $\delta v$, can be expressed by the uncertainty of $V_A$ and $V_B$ as

$$\begin{cases} \delta u = \dfrac{\sqrt{\left(\cos^2 \beta\right) \cdot \left(\delta V_A\right)^2 + \left(\cos^2 \alpha\right) \cdot \left(\delta V_B\right)^2}}{\left|\sin(\alpha - \beta)\right|} \\[4mm] \delta v = \dfrac{\sqrt{\left(\sin^2 \beta\right) \cdot \left(\delta V_A\right)^2 + \left(\sin^2 \alpha\right) \cdot \left(\delta V_B\right)^2}}{\left|\sin(\alpha - \beta)\right|} \end{cases}$$

(11)

where, $\delta V_A$ and $\delta V_B$ are the uncertainty of $V_A$ and $V_B$. When the instruments used for synchronously scanning measurement have the same measurement uncertainty of LOS velocity, $\delta V_L = \delta V_A = \delta V_B$, wind speed uncertainty $\delta V$ and wind direction uncertainty $\delta\theta$ can be expressed as

$$\delta V = \frac{\sqrt{\left(\cos^2 \alpha + \cos^2 \beta\right)\sin^2 \theta + \left(\sin^2 \alpha + \sin^2 \beta\right)\cos^2 \theta}}{\left|\sin(\alpha - \beta)\right|} \delta V_L$$

(12)

$$\delta\theta = \frac{\sqrt{\left(\sin^2 \alpha + \sin^2 \beta\right)\sin^2 \theta + \left(\cos^2 \alpha + \cos^2 \beta\right)\cos^2 \theta}}{V\left|\sin(\alpha - \beta)\right|} \delta V_L$$

(13)

As can be seen from Eq. (12), the uncertainty of wind speed $\delta V$ is dependent on wind direction and the position of measured point, but is independent of wind speed. Eq. (13) indicates the uncertainty of wind direction $\delta\theta$ is dependent on wind direction and the position of measured point, and is inversely proportional to wind speed. The item $|\sin(\alpha–\beta)|$ in both Eq. (12)

20    and Eq. (13) is determined by the position of the measuring point and can be defined as a spatial factor. The spatial factor $|\sin(\alpha–\beta)|$ approaches zero as the measuring point tends to the line through instruments A and B. In this case, both $\delta V$ and $\delta\theta$

increase rapidly, which results in large measurement error. Therefore, the spatial factor can be considered as a reference standard for quality control.

**3 Coplanar scanning mode**

Since the dual-Doppler method requires at least two independent measurements of LOS velocity for retrieving vector wind in each probing volume, higher measurement efficiency can be obtained when a coplanar scan is performed, in which case the intersection of two instruments' scanning surfaces is a plane rather than a curve. Practically, a coplanar scan can be achieved by performing two RHI scans with same azimuth along the line through two instruments (Hill et al., 2010) or two horizontal PPI scans at the same altitude. However, as shown in Fig. 2(a), when two PPI scans with arbitrary elevation are performed, the intersection is a curve. As a result, if a point does not lie on the intersection curve, the altitude difference of these two PPI scans overpassing this position will induce an error in retrieved vector wind (Newsom et al., 2008).

[Figure]

**(a)**          **(b)**

**Figure. 2. Schematic of scanning surfaces and intersections of (a) two PPI scans with arbitrary elevation and (b) two PPI scans in a tilted plane. The black grids show the surfaces of PPI scans, and the red line and red area show the intersection curve and the intersection area, respectively.**

A coplanar scanning mode is proposed to achieve vector wind measurements in a certain plane with high efficiency by adjusting the reference planes of two instruments to a same horizontal or tilted plane according to specified requirement, so that a coplanar scanning measurement can be simply achieved by using two instruments to perform synchronous PPI scans with zero elevation in their reference plane. The schematic of the proposed coplanar scanning mode is shown in Fig. 2(b). Practically, the scanning measurement in a tilted plane can be achieved in two ways: by directly performing scanning measurements in the certain plane without tilting the reference plane of instrument or by accordingly tilting the reference plane of instrument and then simply performing scanning measurement in the reference plane. In the first way, both azimuth and elevation of probing beam are correspondingly changing during the entire scanning process so that they have to be accurately controlled. In this case, a smooth planar scan requires high pointing accuracy and controlling performance of scanning mechanism, especially for a lidar system. In the second way, the reference plane of instrument is required to be adjusted from the horizontal plane to the tilted plane, which is operable for a compact lidar system with the help of an AHRS (Attitude and Heading Reference System). After the corresponding attitude, including heading, pitch and roll angles, is calculated and applied, only a simple PPI in the reference plane with zero elevation is need to be performed. In this case,

since the complexity of scanning measurement is significantly reduced, there is no particular requirement in the aspect of instrument mechanism. The strategy of instrument arrangement and the procedure of attitude calculation are described in detail.

For the purpose of simplifying the description and calculating the attitude that needs to be adjusted, both geographic coordinate system and instrument coordinate system are defined. In the geographic coordinate system, $X$, $Y$ and $Z$ axes point towards North, East and ground, respectively. In instrument coordinate system, the front, right side and underside of instrument are defined as $X_0$, $Y_0$ and $Z_0$ axes, respectively. The azimuth of transmitting beam $\varphi_0$ is defined as the angle between the projection of transmitting beam on $X_0$-$Y_0$ plane and positive $X_0$ axis. When looking downward, $\varphi_0$ increases in a clockwise direction. The elevation of transmitting beam $\theta_0$ is defined as the angle between transmitting beam and $X_0$-$Y_0$ plane. The direction of transmitting beam in the instrument coordinate system $\boldsymbol{r_0}$ can be expressed by a unit vector as:

$$\boldsymbol{r_0} = \begin{pmatrix} x_0 \\ y_0 \\ z_0 \end{pmatrix} = \begin{pmatrix} \cos\theta_0 \cos\varphi_0 \\ \cos\theta_0 \sin\varphi_0 \\ -\sin\theta_0 \end{pmatrix}.$$

(14)

The attitude of the instrument can be described by heading, pitch and roll angles which refer to rotations about $Z_0$, $Y_0$ and $X_0$ axes, respectively. The unit vector $\boldsymbol{r}$ indicates the direction of transmitting beam in geographic coordinate system. The relationship of $\boldsymbol{r_0}$ and $\boldsymbol{r}$ is:

$$\boldsymbol{r} = (H_1 H_2 H_3)^{-1} \boldsymbol{r_0}.$$

(15)

where, $H_1$, $H_2$ and $H_3$ are the rotation matrices of roll $\varphi$, pitch $\theta$ and heading $\psi$, respectively, as follows:

$$\begin{cases} H_1 = \begin{pmatrix} 1 & 0 & 0 \\ 0 & \cos\varphi & \sin\varphi \\ 0 & -\sin\varphi & \cos\varphi \end{pmatrix} \\ H_2 = \begin{pmatrix} \cos\theta & 0 & -\sin\theta \\ 0 & 1 & 0 \\ \sin\theta & 0 & \cos\theta \end{pmatrix} \\ H_3 = \begin{pmatrix} \cos\psi & \sin\psi & 0 \\ -\sin\psi & \cos\psi & 0 \\ 0 & 0 & 1 \end{pmatrix} \end{cases}.$$

(16)

Based on principles of geometry, a plane can be defined by three non-collinear points. Therefore, when the positions of two instruments (points $A$ and $B$), and the target (point $C$) are selected, the tilted plane in which the coplanar scanning is performed can be determined. Theoretically, when the reference plane of instrument is coincident with the tilted plane, the heading of instrument can be arbitrary, which means the solution of instrument attitude is not unique. A practical way to set

up the instruments is to set the $Y_0$ axis of two instruments to be coincident. In this case, when the reference planes of two instruments are adjusted to the tilted plane, the attitude of two instruments will be uniquely and the same. Moreover, the attitude of both instruments can be simply derived by setting the transmitting beam to two special directions. When the transmitting beam is in the tilted plane and perpendicular to the line through two instruments, the unit vector in instrument coordinate system $r_0$ is (1, 0, 0) and the corresponding unit vector in geographic coordinate system $r_1$ is $(x_1, y_1, z_1)$. Similarly, when the transmitting beam is along the line through two instruments, the unit vector in instrument coordinate system $r_0$ is (0, 1, 0) and the corresponding unit vector in geographic coordinate system $r_2$ is $(x_2, y_2, z_2)$. By the way, when the altitude of points $A$ and $B$ are approximatly same, $r_2$ is $(x_2, y_2, 0)$. Both $(x_1, y_1, z_1)$ and $(x_2, y_2, z_2)$ can be derived from the coordinates of points $A$, $B$ and $C$. According to the relationship of $r_0$ and $r$ shown in Eq. (15), $r_1$ and $r_2$ can be derived as

$$
\begin{cases}
r_1 = \begin{pmatrix} x_1 \\ y_1 \\ z_1 \end{pmatrix} = \begin{pmatrix} \cos\psi\cos\theta \\ \sin\psi\cos\theta \\ -\sin\theta \end{pmatrix} \\
r_2 = \begin{pmatrix} x_2 \\ y_2 \\ z_2 \end{pmatrix} = \begin{pmatrix} \cos\psi\sin\theta\sin\varphi - \sin\psi\cos\varphi \\ \sin\psi\sin\theta\sin\varphi + \cos\psi\cos\varphi \\ \cos\theta\sin\varphi \end{pmatrix}
\end{cases}
\tag{17}
$$

Finally, the heading $\psi$, pitch $\theta$ and roll $\varphi$ angles of two instruments can be obtained as

$$
\begin{cases}
\psi = \arctan\left(y_1/x_1\right) \\
\theta = -\arcsin z_1 \\
\varphi = \arcsin\left(z_2/\cos\theta\right)
\end{cases}
\tag{18}
$$

The measurement target (point $C$) can be either a specified point or the center of an interested volume. Once the target is given, the suitable positions of instruments (points $A$ and $B$) can be selected according to the terrain and conditions of the site. Usually, the background wind field, detection range of instruments should be taken into consideration as well to reduce measurement uncertainty. At last, the attitude that needs to be adjusted can be derived accordingly. Moreover, when $z_2$ is zero, roll $\varphi$ of two instruments is zero according to Eq. (18). In this case, only heading $\psi$ and pitch $\theta$ angles of two instruments are needed to adjuste and pitch $\theta$ is the angle of the titlted plane.

**4 Description of the experiment**

A dual-lidar experiment for measuring wind turbine wakes was performed on December 15, 2014, with cloudy weather condition and westerly to northwesterly wind at the Jiangsu Rudong intertidal wind farm ( 32 °31'17" N, 121 °10'32" E). Two lidars were deployed on the embankment with the intertidal zone at the top right of the line through two lidars and the coast at the bottom of Fig. 3. The tilted plane scanning mode was adopted for turbine wakes observation with elevation denoted by

the gray contour lines in Fig. 3. The 2-MW wind turbine T1 had a 100-m rotor diamter and a hub height of 80m. The hub height and rotor diamer of the 1.5-MW wind turbine T5, T6 and T7 were 80 m and 93 m, respectively. Two gasoline engine generators were deployed for power supply. Sea Surface Temperature (SST) from European Centre For Medium Range Weather Forecasts (ECMWF) showed that the sea surface temperature was appximatly 10 ° which was at least 4 ° higher than the air temperature (-1 °–6 ° at local weather station http://lishi.tianqi.com/rudong/201412.html). To obtain the rough time when tide began to rise and reached highest point, the tidal level data was achieved from the nearest Yangkou Harbour (http://www.chinaports.com/chaoxi), located southeast with a distance of approximately 18 km. Data from the harbour showed that tide began to rise after 1204 local time (LT) with tide height of 222 cm and reached the maximum value of 398 cm at 1811 LT. Accordingly, the tide at the experiment was supposed to be rising from 1300LT to 1700LT.

Two pulsed coherent Doppler lidars (WindPrint S4000) operated are from Seaglet Environmental Technology and modified by Ocean University of China (OUC), with AHRS and GPS (Global Position System) modules to record attitude and position information. Laser pulse width was tuned to 100 ns, corresponding to spatial resolution of 15m in the radial direction. This system has pulse repetition rates of 10 kHz and scanning speed of up to 55 %s, which enables LOS velocity to be measured with temporal resolution of up to 0.25 s. Besides, access to high spatial resolution of up to 15 m during scanning measurements could be provided by the tunable pulse width of 100–400 ns and beam pointing accuracy of 0.1 °. The component parameters of two systems are given in (Wu et al., 2016). The effective operating range of both lidars were approximately 1.5 km and 2 km, respectively, owing to different optical efficiency between two lidars, which are denoted by shadows as shown in Fig. 1. The contribution of vertical wind component to the measured LOS velocity could be neglected due to the small elevation of laser beams. Wind vector at a certain point could be precisely retrieved from two non-collinear LOS velocities. Therefore, two-dimension vector winds can be obtained by applying this method to the overlap of two scanning ranges.

[Figure]

**Figure 3: Map of the experiment at an intertidal wind farm of Rudong in Jiangsu province of China with the position of wind turbines (denoted by red dots), two scanning lidar (denoted by balloon A and B), the measurement area (denoted by red and green shadows). The gray lines are contours of the tilted measurement plane.**

5   The heights of two instruments on the embankment are approximately same and the roll angle of them can be tuned to zero degree. Heading and pitch angle of two lidars were 28.5 ° and 4 °, respectively. Then, the angle of both tilted planes was 4 °, adapted to various wind directions and the requirement of obtaining as many wake measurements as possible. In this case, altitude of the tilted plane at wind turbine T1 was about 45m in the low part of wake (wind turbine rotor disk spans a height of 30–130 m). That was favorable to observe the wake in the far wake region because wake decays much more rapidly above

10   the hub height than below the corresponding owing to ground effect (Aitken et al., 2014; Elliott and Barnard, 1990). Each scanning took approximately 90 s as the azimuth ranges of two systems were set to 180 ° with the angular velocity of 2 °s⁻¹ programmed in a script in advance. In practice, the azimuth range could be reduce to 120 °, which further reduced the period of each scanning. The effective measurement time ranged from 1020 LT to 1638 LT, which enabled approximately 220 data sets to be collected. Results contaminated by upwind turbines and severe fluctuation of ambient wind field were screened out

15   and the rest were analysed in what follows. Quality control procedures were applied to LOS velocity by setting threshold of Signal-to-Noise Ratio (SNR) and effective detection range of 1.5 km for lidar A and 2 km for lidar B, and applied to vector wind by setting the spatial factor $|\sin(\alpha-\beta)| \geq 0.35$.

**5 Results**

**5.1 Wake evolution with rising tide**

As mentioned in section 1, turbine wake length $L$ is defined as the distance where velocity deficit $\delta(x)$ drops down to 10%. Ambient velocity $U_{ref}(x)$ is the upwind velocity (Käsler et al., 2010) or the lateral velocity outside of the wake region (Smalikho et al., 2013). When the wind turbine operates susceptible to others, the ambient velocity is more turbulent and slightly smaller than that in free stream. In this case, the lateral velocity outside of turbine wakes should be taken as ambient velocity. While, $U_{wake}(x)$ is the wake velocity along wake orientation. A case of retgrieved field is shown in Fig. 4, in which the spatial resolution of wind speed (denoted by colous) is 15m and the corresponding of wind direction (indicated by arrows) is 45m. The wake induced by wind turbine T1 appeared as cooler colours clearly in the downwind direction. Incremental cross sections are plotted as black rectangles and red squares in Fig. 4 to extract ambient and wake velocities shown in Fig. 5 as black crosses and red squares, respectively. And top 5% velocities in each cross section were selected as ambient velocity $U_{ref}(x)$ and denoted as black points. The ambient and wake velocities were then averaged by 0.5 D axial bins as black and red triangles. Consequently, the velocity deficit $\delta(x)$ against the distance downwind was straightforward to be calculated from Eq.(1) as shown in Fig. 6. The velocity deficit did not apparently reduce when x ≤ 6 D due to the increasing altitude along the wake direction, from approximately 45 m to 80 m, resulting in getting close to hub height (relatively higher velocity deficit in vertical direction). By the way, the base of wind turbine T1 located in the intertidal zone was lower than both two lidars, resulting in that the hub height of T1 is slightly lower than 80 m. However when the downstream distance was beyond 6 D, the velocity deficit decreased sharply , partly due to keeping away from hub height.

[Figure]

**Figure 4: Retrieved vector wind field at intertidal wind farm in China from 1327 LT to 1330 LT on December 15, 2014. The spatial resolution of wind speed is 15m and the corresponding of wind direction is 45m. Red squares and black points indicate two lidars and wind turbine generators, respectively. The gray lines are contours of the titled measurement plane. Velocities in the incremental cross-sections (denoted by red squares and black rectangles) could be extracted to calculate ambient and wake velocities.**

[Figure]

**Figure 5: Wind velocities in the incremental cross-sections (as indicated by red squares and black rectangles) are plotted against the distance downwind as red squares, black crosses. The black points are top 5% of black crosses in each cross-section and used as ambient velocities. The red and black triangles are calculated by 0.5 D rang bins.**

[Figure]

**Figure 6: Variation of wake velocity deficit with downstream distance (in the form of rotor diameter D) is indicated by blue points and wake length L is calculated as the distance where velocity deficit drops down to 10%.**

The maximum wind speed and mean wind direction in the altitude of 60 m, 70 m and 80 m were then averaged by 30 min temporal bins as ambient wind speeds and directions shown in Fig. 7(a) and (b) respectively. The turbulence intensity, $I_u = {\sigma_u}/{U}$, could be calculated as the dividing standard deviation of velocity $\sigma_u$ by the mean of velocity $U$ and shown in Fig. 7(c). The variation of wake length L with rising tide could be obtained by applying the method described previously to the wind field results which have clearly visual feature as same as Fig. 3 with a total number of 26, and is shown in Fig. 7(d). Recall that tide rose from approximately 1204LT and reached the maximum level at about 1811LT. At 1525 LT, the

intertidal zone was completely covered by tidewater exactly according to experimental records. The tide rising period ranging from 1230 LT to 1700 LT was divided into two stages. The wake of wind turbine T1 in this period located in the intertidal zone judging from that wind direction veered from southwest (about 220°) at 1200 LT to west (270°) at 1413 LT and to northwest (295°) at 1700 LT.

5    In stage 1 (from 1230 LT to 1525 LT), the underlying surface transformed from mud to sea water and the mean wake length was about 7D with southwesterly wind until 1410 LT. Subsequently, wake length increased to about 10 D at 1415 LT and 11.7 D at 1450 LT with no obvious fluctuation of wind speed ($9.0$ ms$^{-1}$–$8.8$ ms$^{-1}$) and turbulence intensity (7%–10%) at 80 m height. Typical roughness of flattish ground and ocean surface is approximately 0.03 m and <0.001 m (wind speed < 20 ms$^{-1}$), respectively (Emeis, 2012). Accordingly, this wake length increase was primarily caused by the influence of surface
10   roughness transition.

In stage 2 (from 1525 LT to 1638LT), the hub height relative to the underlying sea surface decreased due to the rising tide with wind speed ranging from $8.4$ ms$^{-1}$ to $9.5$ ms$^{-1}$, wind direction veering from approximately 285° to 290° and turbulence intensity fluctuating around 10%. The obviously declining to 7.8 D and 8.4 D appeared at 1542 LT and 1546 LT with wake orientation veering from 93° to 115°, respectively, presumably due to that the yaw angle of the wind turbine could not be
15   tuned immediately to be aligned with ambient wind direction. Despite all this, wake length increased obviously to 15.6 D at 1554 LT with wind speed, wind direction and turbulence intensity of 9 ms$^{-1}$, 285° and approximately 10%, respectively, at 80 m height. In this case, underlying surface roughness was not supposed to vary significantly because of no obviously wind speed fluctuating. Therefore, this wake growth was mainly attributed to the tide rising leading to decreased hub height or equivalently enhanced "ground effect", which weakened wake decay in far wake region. It should be pointed out that the the
20   higher temperature of the sea surface (approximately 10°) than the air (less than 6°) may resulte in surface turbulence which subsequently reduced the wake length by stronger dissipating effect than in stable boundary layer. The unstable air flow possiblely leaded to the wake length reducing to 14.5 D and 13.5 D at 1614 LT and 1616 LT, respectively. However, the wake lenth still increased to 18D at 1638LT due to rising sea level.

To sum up, the case preliminarily showed that, at least qualitatively, wake length on that day increased in tide rising period
25   resulting from underlying surface roughness transition, rising sea level.

[Figure]

**Figure 7: Variation of ambient wind speed (a), wind direction (b), turbulence intensity (c) and wake length and orientation (d) with rising tide from 1200 LT to 1700 LT. The rising tide period is divided into three stages as shown by different shadows in (d). The wind speed (a), wind direction (b) and turbulence intensity (c) are given in the height of 60 m, 70 m and 80 m.**

5   **5.2 Wake meandering**

Wake meandering, defined as a random oscillation, is generated and driven when the turbulence length scales are larger than the wake width (España et al., 2011) based on the basic wake meandering model (Bingöl et al., 2010; Larsen et al., 2008). It can be used to evalute power production, wind turbine loading (Chamorro and Porté-Agel, 2010; Larsen et al., 2008). Significance of wake meandering research lies in its potential for application in optimization of wind farm topology and

10   operation as well as in the optimization of wind turbines for wind farm applications (Larsen et al., 2008). It is shown that wake meandering is greater at higher ambient turbulence intensity conditions (Abkar and Porté-Agel, 2015; Bingöl et al., 2010; Chamorro and Porté-Agel, 2010) and lidar measuring technique could further investigate the fundamental assumption behind the present meandering theory (Larsen et al., 2008).

Since the cross-section wind profile has the distribution of single or double Gaussian shape, wake centre, and width could be

15   subsequently deduced by the least square fit method with single or double Gaussian curve described in section 2. As shown in Fig. 8, wind speed in cross-section as a function of relative distance to wake centre was fitted by the least square method with single and double Gaussian distribution denoted by red and black curves, respectively. While the wake centre was obtained from Eq.(1) and Eq.(2) by a preliminary fit process. The fit was performed iteratively in an attempt to estimate the upper and lower range of variables. However, only one fitted curve with minimum fitting RMSE (root-mean-square error)

20   was adopted to calculate wake centre and width. Then, wake centre line and wake length downwind behind the turbine could

be derived by applying this method to the incremental wake sections. As shown in Fig. 9, wake centre and width are denoted by black lines and points, and the meandering effect could be seen clearly downwind of turbine T1 with approximately 16 D wake length.

As shown in Fig. 10, wake width reduced from 2 D at the distance of $x$=0.5 D to 1.4 D at $x$=5 D with little variation of wake centre height (about 45 m in Fig. 11), which was most likely relative to the fact that wake centre shifts upward in vertical location (Aitken et al., 2014; Papadopoulos et al., 1995). After 5D, wake meandering was obvious and the height of wake centre would rise due to the result observed in the tilted plane, which could be seen clearly in Fig. 10 that the height of wake centre in tilted plane rose at the distance of $x$ > 5D. Meanwhile, observed wake width grew to about 4.2 D at approximately 10.5D downwind of T1 arising from that wake expands larger with downwind distance (Aitken et al., 2014). However, wake width fluctuated obviously ranging from 4.2 D to 1.5 D in far wake region as shown in Fig. 10. The main reasons were the turbulence diffusing effect resulting in less precise detection, inherent variability of the wake, as well as the meandering effect (Aitken et al., 2014). It should be pointed out that the observed wake width was unavoidably underestimated compared with the horizontal width across wake centre at hub height because the height of observed wake region (from about 45 m to 65 m) was below hub height (80 m).

[Figure]

**Figure 8: Variation of the wind speed (denoted by blue points)in a transect across wind turbine wake as a function of the distance from the center of the wake, are fitted by single Gaussian (red curve) and double Gaussian (black dash curve) curves, respectively.**

[Figure]

**Figure 9: Retrieved wind speed (with spatial resolution of 15 m) by two synchronously scanning lidars (denoted by red squares) around wind turbines (denoted by black points). Centre line and width (in the titled plane) of the wake behind wind generator T1 are denoted by black points and lines, respectively.**

[Figure]

**Figure 10: Variation of wake width in the titled plane with downstream distance from wind turbine generator T1**

[Figure]

**Figure 11: Height of wake boundary (up and down parts in tilted plane) and centreline observed in tilted plane.**

Statistics of the wake centrelines is shown in Fig. 12 and the displacement to lateral distance against increasing wake axial distance is irregular with assumption of Gaussian frequency distribution in horizontal lateral direction $y$ (Högström et al., 1988):

$$f = A\exp(-\frac{y^2}{2\sigma_c^2}) ,\tag{6}$$

while, $f$ has the normalizing requirement,

$$\int_{-\infty}^{\infty} f(y)\, \mathrm{d}y = 1 ,\tag{7}$$

and $\sigma_c$ is the standard deviation of Gaussian frequency distribution with linear increase at longitudinal distance $x \leq 1\,km$ downstream of the wind turbine:

$$\sigma_c = kx ,\tag{8}$$

where, $k$ is the slope value with 0.053 in (Högström et al., 1988). Subsequently, the standard deviation of wake centreline displacement was calculated and presented in Fig. 13 as black squares. The linear relationship between longitudinal distance $x$ and $\sigma_c$ was calculated by linear fitting and indicated as blue line in Fig. 13 with slope value of 0.057, which was slightly larger than 0.053 presumably due to stronger mesoscale wind fluctuations (Högström et al., 1988). However when longitudinal distance exceeded 10 D, the nonlinear relationship was obvious that $\sigma_c$ has the similarly exponential increase and expressed as red curve in Fig. 13. That would be the main reason why the wake lengths observed by in situ measurements were less than by other remote sensing instruments.

[Figure]

**Figure 12: Statistics of wake centrelines displacement in lateral direction along wake axial distance**

[Figure]

Figure 13: Standard deviation of centreline position as a function of axial distance in wake direction, the black squares are calculated from Figure 12, the blue line indicates the linear fit at the distance of <1 km (10 D). The red curve represents exponential increase in the range of <14 D.

**6 Conclusions**

In this study, we analyze wind turbine wake in the intertidal zone analysis based on the dual-Doppler method with a coplanar plane scanning strategy. The conclusions drawn are in what follows.

The dual-lidar techonology with designed coplanar scanning mode could be properly applied in the situation of variable wind direction compared with traditional PPI or RHI scanning mode, in which the lidar-turbine line should be strictly aligned or roughly same with the turbine wake. Besides, wake analysis could be directly based on retrieved wind field, which is more convenient than LOS velocity or projection along wake direction utilizing only one lidar system. The inclination angle of the tilted plane was tuned to 4 °. In this case the altitude of the wind turbine T1 was approximately 45 m and below hub height of 80 m, which was favorable to observe the far wake region to some extent. The pointing accuracy of the lidar beam scanner and the temporal difference of two LOS velocities at the same points should be further taken into account, which can be optimized by reducing angular velocity and scanning a specific wake region, respectively. However, the ideal mounting location of lidar is on the nacelles, in which case wake structure in horizontal and vertical direction at hub height could be scanned by PPI with zero elevation angle and RHI mode transecting the wake centreline due to lidar yawing along with the wind turbine. This method is an improvement for ground-based measurmenets.

The case analysis showed that wake length on that day evolved with rising tide at least qualitatively due to the impact of underlying surface roughness transition and rising sea level on the wake. Variation of wind speed and turbulence is not obvious, but the wind direction veered from southwest to northwest. The observation range is limited when wind belowed from southwest. However, there was no case of wake lenth beyond the observation range in all measurments. It is necessary to carry out on several periods of wake measurements for each set of turbines with similar characteristics (hub-height, rotor diameter, wake fetch etc). Wake meandering case was also analysed based on the characteristic of the cross-section velocity distribution.

The previous method of velocity deficit calculation might be affected by the meandering especially in the far wake region as the selected wake regions deviate from the wake centreline. Nevertheless, as the wake propagated downstream and was mixed by small-scale ambient turbulence, it was difficult to distinguish the centreline of sufficiently dissipated wake. Moreover, the wake meandering introduced a new problem of whether the wake length was calculated by the wake centreline length or the length from wake centreline to the wind turbine. Besides, wake meandering is a random oscillation as mentioned above, which seemed to have no relationship with rising tide. By considering all these, the severe meandering wakes were ruled out in the wake length calculation for simplification.

Both sides of the observed ambient velocities outside of the wake were different due to the tilted plane resulting in different heights, which would cause asymmetry double Gaussian curve and further bring large bias of wake centre and width. As a result, the length of the cross-section should be narrow enough. However to guarantee the fitting precision, more data points means longer cross-section. Accordingly, the adopted length of cross-section was a tradeoffs between both requirements.

In summary, wake behaviour could be properly observed based on the dual-lidar method for its feasibility in variety wind direction. But the more ideal mounting location of lidar is on the nacelles. Besides, further field experiments shall be performed to quantify the dependence of the wake behaviour (velocity deficit, wake length, wake boundary and wake centreline) on the atmospheric condition (wind speed, turbulence, surface roughness and atmospheric stability) combined with turbine model, and ultimately improve the prediction of wind power harvesting.

**Acknowledgement**

We thank our colleagues for their kind support during the field experiments and results discussion, including Quanfeng Zhuang, Guining Wang and Xiaoqing Yu from Ocean University of China (OUC) for the preparing and conducting the experiment; Yilin Qi and Jie Bai from Seaglet Environment Technology for preparing and operating the lidar in the cold and windy wind farm. This work was partly supported by the National Natural Science Foundation of China (NSFC) under grant 41471309 and 41375016.

---

## Author Comment (AC2) · 8 May 2017

We sincerely thank you for your constructive advice and kindness. We have modified the revision and answered every specific comment in the supplement, in which the new revision is supplied.

Please also note the supplement to this comment:
http://www.atmos-meas-tech-discuss.net/amt-2017-23/amt-2017-23-AC2-supplement.zip